# A Systematic Review and Meta-Analysis Investigating the Efficacy of Various Psychedelic Drugs for the Treatment of Substance Use Disorder

**DOI:** 10.3390/healthcare13212668

**Published:** 2025-10-23

**Authors:** Eve E. Keighley, Eid Abo Hamza, Dalia A. Bedewy, Shahed Nalla, Ahmed A. Moustafa

**Affiliations:** 1School of Psychology, Faculty of Society and Design, Bond University, Gold Coast, QLD 4226, Australia; evekeighley@hotmail.com (E.E.K.); ahmed.moustafa@bond.edu.au (A.A.M.); 2College of Arts, Humanities and Social Sciences, University of Sharjah, Sharjah P.O. Box 27272, United Arab Emirates; eabohamza@sharjah.ac.ae; 3Department of Psychology, College of Humanities and Sciences, Ajman University, Ajman P.O. Box 50804, United Arab Emirates; 4Faculty of Education, Tanta University, Tanta 31511, Egypt; 5Department of Human Anatomy and Physiology, The Faculty of Health Sciences, University of Johannesburg, Auckland Park P.O. Box 524, South Africa; shahedn@uj.ac.za; 6Centre for Data Analytics, Bond University, Gold Coast, QLD 4226, Australia

**Keywords:** substance misuse, substance use disorder, psychedelic treatment, ibogaine, psilocybin

## Abstract

**Objectives:** This study investigates psychedelic drugs to treat substance use disorder (SUD). Researchers have recently begun conducting clinical trials of psychedelic treatment for SUD. The current meta-analysis investigates the extent of efficacy in alleviating SM behaviours (P) using psychedelic therapy (I), concurrent with determining which psychedelic enables the greatest effect (C) as a treatment tool for reducing SUD (O). **Methods:** The inclusion criteria in this study include evaluating the efficacy of LSD, psilocybin, ketamine, or ibogaine in human beings with an SUD. The exclusion criteria include studies on rodents, patients with schizophrenia, case studies, incomplete or ongoing trials, and studies with insufficient quantitative data. The search criteria obtained 1278 articles, acquired through PubMed and PsycINFO. After excluding literature, 30 papers were kept in the final meta-analysis. A random-effects model analysis was applied to investigate individual psychedelic interventions, with a corresponding combined psychedelic intervention analysis. **Results:** The results favoured psychedelics as an SM treatment, with ibogaine evidencing the most prominent. We also found a non-significant difference between the effectiveness of psychedelic treatment paired with psychotherapy and psychedelic treatment alone. This study aims to contribute knowledge to future clinical research on the psychedelic treatment of SUD.

## 1. Introduction

Substance use disorders (SUD) can be classified as psychiatric disorders that commonly coincide with dysfunctional and destructive behaviours [1,2]. The *Diagnostic and Statistical Manual of Mental Disorders* (5th Ed: American Psychiatric Association [APA], 2013) [3] states that SUD is defined as the use of a particular substance (e.g., legal or illegal drugs) despite subsequent destructive cognitive, behavioural, and physiological symptoms. According to the United Nations Office on Drugs and Crime [4], 296 million people used drugs in 2021, with an estimated 39.5 million people worldwide currently affected by an SUD, of which half a million deaths are annually attributed to substance misuse [5].

Cannabis remains the most used illicit substance, with an estimated 219 million users in 2021 (UNODC, 2023). Alcohol, a psychoactive substance, causes around three million deaths annually, with 13.5% of fatalities seen in those aged 20 to 39, and is attributable to 5.1% of the global burden of disease and injury [5]. In 2018, 14.8 million SUDs were alcohol-related [6]. In 2021 alone, roughly 22 million individuals aged 15 to 64 reportedly used cocaine the previous year, with Oceania (including the subregion of Australia and New Zealand) attaining a usage prevalence of 2.7%. An estimated 60 million people were reported using opioids and amphetamine use among the general population was estimated at 36 million, with the highest majority in North America [4,5]. Crystalline methamphetamine (i.e., crystal meth) is the most predominant amphetamine in most countries, including Australia. According to reports from 2021, 34% of people seeking treatment for SM in Australia and New Zealand were under 25 years old [7].

### 1.1. Psychedelics

*Psychedelics*, also known as “hallucinogens”, are psychoactive substances that can alter human perception, cognition, mood, and thought processes [8,9]. Between 1950 and 1960, researchers conducted clinical trials using psychedelics and observed promising results in patients who sustained mood disorders and SUDs [8,10,11].

Over the past decade, clinical trials have begun re-exploring the effects of Lysergic Acid Diethylamide (LSD), psilocybin, ketamine, ibogaine, and other compounds for treating mental health disorders and SUDs [12,13]. Recent reviews [14,15] further support this resurgence, noting promising clinical outcomes across alcohol, opioid, and stimulant use disorders, while emphasising the urgent need for standardised methodologies and rigorous trial designs.

Individuals who consumed psychedelics displayed remarkable effects, including long-lasting personality alterations, behavioural changes, reductions in cravings, and symptom relief, with over a thousand papers documenting the treatment for nearly 40,000 individuals during this period [16,17]. Unfortunately, research studies for psychedelics were short-lived as the ban on these mind-altering drugs occurred in the early 1970s. Over the past decade, clinical trials have begun re-exploring the effects of Lysergic Acid Diethylamide (LSD), psilocybin, ketamine, ibogaine, and other compounds for treating mental health disorders and SUDs [18]. Below, we briefly discuss major psychedelics in some detail.

### 1.2. Psilocybin

Psilocybin is a compound found in over 100 species of psychedelic mushrooms and has long-lasting positive effects on psychiatric disorders and SUDs. Psilocybin is classified as a serotonergic hallucinogen and is an agonist on the brain’s 5-hydroxytryptamine 2A receptors [10,19,20,21]. Psilocybin has been attributed its non-addictive, anti-depressant, and anti-anxiety nature in clinical and non-clinical settings [10,22,23]. The outcome from clinical studies yields significant results about psilocybin efficacy to reduce addiction. A recent systematic review by van der Meer et al. [11] synthesised decades of psilocybin research, including both contemporary and overlooked historical trials, and found consistent reductions in substance use behaviours alongside improvements in psychosocial functioning.

One study in 2015 prescribed psilocybin to alcohol-dependent patients [24]. Ten individuals received oral administration of psilocybin over one or two sessions concurrently with motivational interviewing. Results exhibited a significant reduction in the participants’ alcohol consumption at the 36-week follow-up. The researchers found that participants only maintained abstinence once psilocybin had been administered; individuals had received therapy for four weeks prior with no observed changes. Therefore, it was indicated that behavioural therapy alone is not enough to alleviate SM. An ongoing pilot study [25] has begun preliminary trials using psilocybin treatment for cocaine use disorder. Thus far, a randomised control trial including 10 participants (with a proposed *N* of 40) either received a dose of psilocybin or a placebo comparator. Results showed a significant reduction in cocaine use at the six-month follow-up for those in the psilocybin treatment group compared to the control group.

### 1.3. LSD

Lysergic Acid Diethylamide has low toxicity and abuse potential and is a serotonergic hallucinogen that works as a 5-HT2A agonist [10,23]. Researchers used LSD to treat alcoholism throughout the middle to late 20th century. A meta-analysis conducted in 2012 by [26] assessed the meaningful effects of LSD in aid for alcoholism from six randomised trials. Across six studies, 325 participants were randomly assigned to receive a dose of LSD, and 211 participants were assigned to a control condition. Results demonstrated the effectiveness of LSD on alcohol misuse for up to six months [27,28,29,30,31,32,33].

A research study conducted in the 1970s [34] administered LSD to reduce substance misuse in heroin addicts. Seventy-eight inmates from a correctional facility were randomly assigned to a treatment (*n* = 37) or control group (*n* = 37). The treatment group was administered one dose of LSD-assisted psychotherapy and the control group were undertaking weekly group psychotherapy with no psychedelic administration. The LSD group displayed significantly higher abstinence outcomes at both the 0–6-month and 7–12-month follow-up compared to the control group [27].

Individuals reported taking a moderate to high dose of LSD (43%), psilocybin (29%), or other (28%). The authors of Ref. [9] argue that the therapeutic benefit of such psychedelic encounters may be mediated by the intensity of mystical experiences, which, in turn, predicts sustained reductions in addictive behaviours. Moreover, an anonymous online survey by Garcia-Romeu et al. 2020 [35] assessed individuals’ reduced SM following psychedelic intake. Of the 444 respondents, 96% of individuals met the criteria for an SUD and 79% for a severe SUD. Individuals reported taking a moderate to high dose of LSD (43%), psilocybin (29%), or other (28%). Following psychedelic encounters, only 27% met the criteria for an SUD. The most significant reductions in SM were associated with those who reported a highly personal experience.

### 1.4. Ketamine

Ketamine is commonly used as an anaesthetic but can produce psychedelic effects at a subanaesthetic dose; ketamine acts as an N-methyl-d-aspartate receptor (NMDAR) antagonist. Unlike LSD and psilocybin, ketamine directly interacts with glutamate receptors implicated during the development of an SUD [14,36,37,38]. Jones et al., 2022 [39] conducted a systematic review, assessing seven studies to treat SUDs with ketamine (two focused on alcohol, two on cocaine, and three on opioids). Results demonstrated an improvement in abstinence for up to two years in alcohol and opioid addicts, and cocaine users reported a decrease in cravings and cocaine use. Researchers are optimistic regarding ketamine as a treatment option due to its promising and successful outcomes, but broader investigations have been suggested.

### 1.5. Ibogaine

Ibogaine is a psychoactive alkaloid found in Central Africa. Ibogaine produces intense psychedelic states, often focusing on subjective life events where users do not return to ordinary consciousness for 72 h. Ibogaine has been known to reduce craving and withdrawal symptoms in opioid and cocaine users [14,40,41]. Davis et al. [42] evaluated the effectiveness of ibogaine for problematic opioid consumption. Using an online survey, 88 opioid-dependent patients reported their subjective experience after ibogaine treatment. In total, 80% of individuals indicated that ibogaine drastically reduced opioid withdrawal symptoms, 50% reported diminished opioid cravings, and 30% reported total abstinence for up to two years. The authors suggested that ibogaine can produce long-term positive outcomes for opioid-addicted patients but suggest further research using rigorous longitudinal and controlled designs.

Furthermore, Brown and Alper, 2018 [43] used ibogaine to treat 30 opioid-dependant subjects, endeavouring to study the drug’s effects on withdrawal outcomes following opioid detoxification. At the one-month post-treatment follow up, 50% (*n* = 15) of subjects reported sustained abstinence from opioid use during the previous 30 days. Researchers concluded that maximised efficacy of the outcome was highest at the one-month post-treatment compared to 3 to 12 months post-treatment, thus determining that ibogaine was associated with substantive efficacious outcomes for withdrawal symptoms in those who were unsuccessful with opponent treatments, suggesting that ibogaine may serve as a useful prototype for the discovery and development of novel medication for addiction [15,17].

Although the prior literature favours the benefits of psychedelic therapy, Sharma et al. [44] conducted a systematic review investigating the efficacy of psychedelic treatments for patients with SUD and those who fell below diagnostic standards. Researchers measured abstinence, substance use, craving, and withdrawal, which demonstrated positive results. However, the researchers still highlighted the lack of evidence that supports psychedelics as a successful treatment for SM. According to the researchers, the subjective experience of psychedelic-assisted therapy boosted self-awareness, insight, and confidence in three studies but stated that there is currently insufficient data to establish the efficacy of any specific psychedelic in treating substance use disorders or misuse. Authors also mentioned that additional research employing rigorous evaluation methodologies, larger sample numbers, and longer-term follow-up will be required to determine a more definitive outcome.

### 1.6. The Current Review

In 2021, clinical trials of psychedelic-assisted therapy were approved in Australia for individuals suffering from mental illnesses. The Therapeutic Goods Administration confirmed that these medicines were safe, well-tolerated by individuals in medical settings and produced statistically significant results [45]. Despite the proposed psychedelics’ varying chemical, biological, and psychological mechanisms, they share the same desired outcome when used as an SM treatment—to reduce maladaptive drug-taking behaviours. Therefore, the current article sought to investigate the efficacy of psilocybin, LSD, ketamine, and ibogaine as a treatment for SUD and SM.

As the previous literature supports the treatment of psychedelics to reduce drug addiction [24,26,34,35,39,40,42], it is expected that the current meta-analysis will result in favour of psychedelics to manifest the abstinence and alleviation of SM. Further, Bogenschutz et al., 2015 [24] suggested that psychotherapy alone is insufficient to reduce SM. Therefore, it is predicted that psychedelic treatment concurrent with psychotherapy will display superior benefits over psychedelic treatment alone. Krebs and Johansen, 2012 [26] found LSD to display promising outcomes up to six months post-treatment. However, Bogenschutz et al., 2015 [24] observed the efficacy of psilocybin for up to 36 weeks post-treatment. Additionally, the current research will explore the variance between psychedelic treatment efficacy regarding the length of treatment and post-outcome follow-up.

## 2. Methods

### 2.1. Primary Searches

A systematic literature search was conducted to collect research articles published in pre-eminent online journal databases. The screening of articles was achieved through the online systematic review tool, Covidence [46]. This was conducted in adherence with the Preferred Reporting Items for Systematic Reviews and Meta-Analysis (PRISMA) guidelines [47]. Preparatory searches were piloted through Google Scholar to refine the scope and terminology prior to running full database searches. Various combinations of keywords were used, such as mentioned in the subsequent section. In the start no filters were applied, and the first couple of results per string were screened by title/abstract to identify recurring terminology and study types. This exercise, undertaken with guidance from Bond University Library, provided clarity on appropriate keywords and informed the final search strings used in PubMed and PsycINFO.

### 2.2. Eligibility Criteria

The compiled studies were primarily screened based on the title and abstract. The eligibility criteria were based on the PICO approach (person, intervention, comparison, outcome; [48]), which aimed to investigate the extent of efficacy in alleviating SM behaviours (P) using psychedelic therapy (I). Concurrent with determining which psychedelic enables the greatest effect (C) as a treatment tool for reducing SM (O). We performed separate analyses for studies that used a treatment vs. control design and studies that used a pre-treatment vs. post-treatment design.

The inclusion criteria were as follows: (a) studies must include the evaluation of efficacy regarding psychedelic (either LSD, psilocybin, ketamine, or ibogaine) use in human beings that sustain an SUD, substance use dependency, SM behaviours, or alleviate substance use cravings and withdrawals; (b) studies can be with or without behavioural therapy; and (c) patients in the studies were not required to attain a formal diagnosis of an SUD, seeing that they had met the DSM-IV criteria, with SM negatively impacting various aspects of their life (i.e., relationships, health, jobs, and finances). Studies evaluating naturalistic psychedelic use were deliberated for removal but included in the final analysis due to restrictions on psychedelic use in clinical settings (i.e., individuals may have used psychedelics on their own accord).

Studies were excluded if the title, abstract, or full text mentioned psychedelic treatment on rodents, patients with schizophrenia, case studies, and incomplete or ongoing trials. Studies not written in English and irrelevant or redundant literature that did not include concurrent psychedelic treatment for SM or any related domain were excluded. Additionally, studies that needed to provide more quantitative data were deemed ineligible. See Figure 1 for the PRISMA flow diagram.

### 2.3. Search Criteria

Literature searches were performed electronically. A comprehensive search of articles was acquired through PubMed and PsycINFO databases from inception to 5 July 2025. The following keywords were used to gather results from PubMed and PsycINFO: (“treatment substance use disorders” OR “treatment drug dependence”) in conjunction with terms like “psychedelics”, “hallucinogens”, “ketamine, “LSD”, “psilocybin”, and “ibogaine”. PsycINFO database was searched using the subsequent terms: (“drug addiction” OR “drug withdrawal” OR “substance use disorder” OR “cannabis use disorder” OR “opiates” OR “opioid use disorder” OR “alcoholism” OR “drug dependence”) AND (“hallucinogenic drugs” OR “psychedelics” OR “psilocybin” OR “ketamine” or “lysergic acid diethylamide” OR “MDMA” OR “ibogaine”) AND (“substance use treatment” OR “treatment” OR “addiction treatment” OR “alcohol treatment”).

Search terms were included for the complete reference as follows: title, abstract, and full text. Inclusion prerequisites encompassed the articles being written in English. As the study of psychedelic treatment has a resurgence, the study sample size was not restrained when determining the selection criteria. There were no date restrictions obligatory for the selected studies. Therefore, the published article dates ranged from 1960 to 2025. As the purpose of the current article was to compare psychedelic efficacy from baseline to post-treatment outcome, the eligibility criteria regarding study design were not constrained. The combined search of the databases resulted in a total of 1284 papers after duplicates were removed. The figure below represents the PRISMA flow diagram, outlining the number of articles that were identified, screened, and included.

### 2.4. Data Analysis

The data from studies was extracted independently by one reviewer. To minimise the risk of error and subjective bias, the extracted data was then independently cross-checked and confirmed by a second reviewer. The following data was reviewed from each trial where applicable: psychedelic type, substance being misused, intervention type, intervention characteristics, assessment of participants, participant characteristics, trial characteristics, information provided to participants, assessment of outcomes, length of treatment, and follow-up time. A formal risk-of-bias assessment was performed for all included studies. Randomised controlled trials (RCTs) were evaluated using the Cochrane Risk of Bias 2 (RoB 2) tool as per a study reported in 2019 [49], which examines five domains: (1) randomisation process, (2) deviations from intended interventions, (3) missing outcome data, (4) measurement of the outcome, and (5) selection of the reported result. Non-randomised and observational studies were assessed using the Risk of Bias in Non-randomized Studies of Interventions (ROBINS-I) tool [49]. Two independent reviewers conducted the assessments, with disagreements resolved through discussion and consensus.

### 2.5. Statistical Analyses Methodology

The current meta-analysis used Meta-Essentials Excel workbook three, “*Differences between independent groups—continuous data*”, and four, “*Differences between dependent groups—continuous data*”, per a study reported in 2017 [50]. The psilocybin, ketamine, and ibogaine treatment studies were analysed using a repeated-measures design. Each study’s control groups were excluded; pre-treatment (baseline) values functioned as a replacement, and post-treatment data was the outcome. The three psychedelic interventions measuring pre- to post-treatment differences were analysed using the Meta-Essentials workbook four (version 1.5), with pre-treatment data entered as the control group when calculating preliminary effect sizes. Due to the LSD studies lacking participants’ baseline data, a comparison between participant pre-treatment and post-treatment efficacy could not be determined. Therefore, LSD studies aim to evaluate the efficacy between treatment and control outcomes, where the outcome data were dichotomised into “improved” or “not improved”. The LSD studies were analysed using the independent-measures Meta-Essentials workbook three.

The included studies were analysed in distinct groups based on their methodological design. Studies on psilocybin, ketamine, and ibogaine were analysed using a pre-treatment to post-treatment design due to the nature of the available data, which often lacked a placebo control group. In contrast, the studies on LSD were exclusively analysed using a treatment vs. control design. This analytical split was necessary to ensure that our meta-analysis did not combine effect sizes derived from fundamentally different types of comparisons, thereby maintaining the methodological integrity and interpretability of the pooled results.

Preliminary effect size calculations were conducted using the Practical Meta-Analysis Effect Size Calculator [51] and Rapid Effect Size Converter for Meta-Analysis, 2020 [52]. The definitive studies used several statistical techniques, transforming the data into a uniform effect size. Group means, standard deviations, *F*-statistics, *t*-tests, chi-square analysis, and frequency values were all transformed into Cohen’s *d* (standardised mean difference). The final analysis required converting Cohen’s *d* values into Pearson’s r value.

The Meta-Essentials workbooks required the following information from individual studies: Cohen’s *d* values, group sample sizes, and Pearson’s *r*. The workbooks produced all the necessary statistical calculations for a meta-analysis. All data from separate trials was converted into a consistent, standardised Hedges *g* effect size, using 95% upper and lower confidence intervals, with forest plots provided. Effect sizes were interpreted as small (*g* = 0.20), medium (*g* = 0.50), or large (*g* = 0.80) as per a study published in 1998 [53], and the alpha level was set at 0.05 to increase statistical power.

A random-effects model was applied to each analysis to estimate average weighted effect sizes due to variability in the study cohorts published in 2018 [54]. Further analyses were assessed through Cochran’s (1952) *Q*-statistic test to indicate the significance of heterogeneity and an *I*^2^ value to quantify the total heterogeneity for each group as per a study conducted in 2002 [55]. If the *Q*-statistic displayed significance, heterogeneity was also significant. The percentage of heterogeneity variance (*I*^2^) that was not due to chance was interpreted as low (25%), moderate (50%), and high (75%), as per a study of 2003 [56]. If heterogeneity was significant and exceeded the low value, a subgroup analysis to identify moderating variables was to be performed to investigate the variance in effect sizes further. There were two pre-planned subgroup analyses; one aimed to investigate the relationship between intervention type and treatment efficacy and the other aimed to investigate the relationship between length of treatment and post-treatment SM reduction.

The asymmetry of effect sizes anticipated by publication bias was assessed by inspecting funnel plots. Edgar’s regression was used to determine the symmetry of the distribution of effect sizes, and a Trim and Fill technique was applied to signify the amount of publication bias [57]. *Z* statistics followed by a one-tailed *p*-value determined the overall effect for each analysis.

## 3. Results

### 3.1. Description of Studies

A total of 30 studies, comprising 1070 participants, were included in the current meta-analysis (refer to Appendix A for Table A1). Participants in the repeated-measures condition (all control groups excluded) received at least one dose of either psilocybin, ketamine, or ibogaine. Participants’ improvement in post-treatment outcome from a study in 2020 was compared with baseline data: percentage of drinking days in a study of 2020 [58], percentage of heavy drinking days in 2015 [24], average cannabis use in 2020 [59], cocaine cravings in 2017 [60], global improvement in 2018 [61], days abstinent in year 2014–2020 [62,63,64], reductions in alcohol consumption in 2019 [65], heroin craving in 2007 and 2002 [38,66], average drinks per day [67], reduction in substance cravings in 2017 and 2018 [68,69], substance withdrawals in 2018 [69], and addiction severity in 2018 [70]. Additionally, recent clinical synthesis work [15] and ketamine-focused reviews [14] were considered for background context, highlighting evolving clinical approaches and their implications for psychedelic-assisted SUD treatment. If high post-outcome scores represented less drug misuse (e.g., more days abstinent), then effect sizes were reversed to ensure all results were interpreted similarly (e.g., reduction in substance use).

The participants from each study either had a formal diagnosis of an SUD or they met the DSM-IV criteria for an SUD. Participants were assessed for psychiatric conditions and deemed ineligible if they had history of schizophrenia or bipolar. The studies outlined a thorough description of the inclusion and exclusion criteria, along with detailed methodological characteristics. Most studies employed a debrief and psychological assessment prior to psychedelic therapy. Abstinence from SM was required, with most psychedelic administration sessions taking place in a comfortable setting within a medical facility. Participants’ psychedelic sessions were accompanied by medical professionals to provide support and ensure the physical and psychological safety of participants. Health assessments were carried out throughout the sessions (i.e., blood pressure and heart monitor). Studies assessed the outcome predominantly through self-report questionnaires, and follow-up assessments were conducted through interviews.

One study evaluated both clinical withdrawal symptoms and addiction severity post-psychedelic infusion; therefore, the study was split into two groups for the analysis [69]. In another trial, individuals were randomly assigned to receive either a high or low dose of ketamine; however, the low-dose group was excluded because only one subject was left at the post-treatment follow-up [66]. Only those who sought ibogaine directly for SM treatment were included [68].

The outcome of participant improvement in the LSD conditions was compared to the control group. In the LSD conditions, good improvement and modest improvement or improved and generally improved outcomes were dichotomised as “improved”, and no improvement or not much improvement was dichotomised as “not improved” [28,71]. Two trials included non-randomised control groups or non-randomised subgroups. Therefore, they were excluded, consisting of a study of 1970 [28,33]. Additionally, one trial included a sub-group of schizophrenic alcoholics and was excluded from the meta-analysis [33].

The current meta-analysis resulted in favour of psychedelics to manifest the abstinence and alleviation of SM. These findings are aligned with recent systematic reviews [9,11], which collectively report meaningful reductions in substance use behaviours and underline the importance of psychological mechanisms such as mystical-type experiences.

### 3.2. Psilocybin

The data from four studies (*N* = 105) assessing the efficacy of psilocybin as a treatment for SM was combined using a random-effects meta-analysis. The studies’ effect sizes ranged from 0.15 to 1.57, with a large, combined Hedges’ g of 1.25 (95% CI [0.93, 1.57]) and a significant overall effect (Z = 7.56, *p* < 0.001), suggesting that psilocybin is effective as a treatment for SUD. Rieser et al. [22] presented a notably smaller effect size (*g* = 0.15), which slightly reduced the pooled estimate compared to earlier analyses. Figure 2 represents a forest plot of individual and combined effect sizes. Bogenschutz et al. [18] hold the greatest weight (54.47%) and Bogenschutz et al., 2015 [24] hold the least (13.32%).

**Heterogeneity.** A non-significant, low heterogeneity was detected (*Q* = 1.08, *p* = 0.582), with an *I*^2^ value explaining 0% variance. As heterogeneity was non-significant, a subgroup analysis identifying moderating factors was not required, as per a study in 2017 [72].

**Publication Bias.** Publication bias was analysed by inspecting a funnel plot (see Figure A1), which depicted an asymmetrical distribution of effect sizes. To quantify publication bias, a Trim and Fill technique was applied, as per a study in 2000 [57]. Results suggested that zero missing studies would correct publication bias. The Eggers regression (study in 1997) displayed a non-significant result (*p* = 0.366), indicating minimal publication bias and asymmetry [73].

### 3.3. LSD

Seven studies (*N* = 555) analysing LSD as an SUD treatment were synthesised using a random-effects meta-analysis model. Studies reported Hedges *g* ranging from −0.10 to 0.69. The meta-analysis depicted a significant overall effect (*Z* = 3.19, *p* = 0.001), with an average overall Hedges *g* of 0.36, 95% CI [0.08, 0.64]. Thus, a significant overall effect suggests that treatment for SUD favours LSD. The forest plot in Figure 3 represents the random effects of LSD; studies with positive values favour the treatment condition and negative values favour the control condition. Ludwig et al., 1969 [30] possessed the most meta-analytical weight (20.04%), with Smart et al., 1966 [32] possessing the least.

**Heterogeneity.** A non-significant (*Q* = 10.75, *p* = 0.096) small to moderate heterogeneity was detected, displaying an *I*^2^ value of 44.20% [56]. Due to a non-significant heterogeneity, subgroup analysis for moderator variables was not required, as per a study of 2017 [72].

**Publication Bias.** Publication bias analyses were performed. Inspection of the funnel plot (see Figure A2) revealed an asymmetrical distribution of effect sizes. A Trim and Fill technique was applied, as per a study of 2000, to quantify publication bias [57], revealing that adding zero studies would correct asymmetry arising from publication bias. The Eggers regression test was non-significant (*p* = 0.467), indicating asymmetry and minimal publication bias.

### 3.4. Ketamine

Nine studies (*N* = 188) assessing ketamine as a treatment for SUD were analysed using a random-effects meta-analysis, with individual studies’ effect sizes ranging from 0.47 to 2.69. A significant overall effect was detected (*Z* = 7.17, *p* < 0.001), with a large, combined Hedges *g* of 1.66 (95% CI [1.12 to 2.19]; [53], a study of 1998). Thus, ketamine possesses a significant influence as a treatment for SUD. A forest plot depicted below (see Figure 4) reveals that Krupitsky et al., 2002 [66] hold the most weight (12.33%) and Azhari et al., 2020 [59] hold the least (9.32%).

**Heterogeneity.** Significant heterogeneity was detected (*Q* = 43.72, *p* < 0.001), with an *I*^2^ value of 81.70% of the variation among the results. A study suggests that the *I*^2^ value can introduce bias regarding heterogeneity when the number of studies in a meta-analysis is small, recommending that a subgroup analysis can only be performed if there is an adequate study size (*k* > 10); a subgroup analysis with less than 10 studies is likely to be imprecise [75]. Due to the limited number of studies, further analyses were redundant and, therefore, were not performed.

**Publication Bias.** Upon inspection of the funnel plot (see Figure A3), a moderately symmetrical distribution of the effect sizes was present. An Eggers regression [73] depicted non-significant results (*p* = 0.398). The Trim and Fill technique [57] suggested that adding zero studies would correct asymmetry resulting from publication bias.

### 3.5. Ibogaine

The data from six studies (*N* = 240) assessing ibogaine as a treatment for SM were computed using a random-effect model. Individual effect sizes ranged from 1.20 to 3.23, with a significant overall effect (*Z* = 6.42, *p* < 0.001) and a combined large Hedges *g* of 2.00, 95% CI [1.20 to 2.80], therefore indicating that ibogaine is effective as a treatment for SM. Figure 5 represents a forest plot displaying the individual and overall effect sizes. Schenberg et al., 2014 [64] have the most weight (18.03%), whereas Noller et al., 2018 [70] have the least (14.52%).

**Heterogeneity.** The ibogaine treatment group demonstrated significantly high heterogeneity (*Q* = 48.69, *p* < 0.001), with an *I*^2^ value of 89.73%. Due to the small number of studies included, the likelihood of an imprecise result is high. Therefore, the performance of a subgroup analysis was redundant [75].

**Publication Bias.** Inspection of the funnel plot (see Figure A4) revealed a non-homogenous distribution of effect sizes. The Eggers regression for ibogaine treatment was significant (*p* = 0.005), indicating asymmetry and positive publication bias contributing to the result. Trim and Fill suggested zero added studies would rectify asymmetry resulting from publication bias [57].

### 3.6. Psilocybin, Ketamine, and Ibogaine Combined

A total of 18 studies (*N* = 515) assessing the efficacy of psilocybin, ketamine, and ibogaine as an SUD treatment were combined. The data were synthesised using a random-effect meta-analysis. The combined analysis was conducted using Meta-Essentials workbook four. Results produced a large, standardised effect size (*g* = 1.73, 95% CI [1.39 to 2.06]), with individual effect sizes ranging from 0.47 to 3.23. The overall effect was significant (*Z* = 10.96, *p* < 0.001).

Therefore, psilocybin, ketamine, and ibogaine display efficacy from baseline to post-intervention when used as a treatment for SUD. Figure 6, displayed below, represents the forest plots for the combined intervention studies. Most of the studies’ confidence intervals overlap, indicating face-value homogeneity of results. The most meta-analytical weight was assigned to ibogaine 65, possibly due to the minor standard error (0.15).

**Heterogeneity.** The combined groups displayed considerable, significant heterogeneity (*Q* = 96.40, *p* < 0.001), with an *I*^2^ value of 82.36%. Therefore, a subgroup analysis was performed to identify moderating factors. A subgroup analysis of time as a moderating factor (post-treatment outcome less than six months vs. post-treatment over six months) was significant (*p* < 0.001, *I*^2^ = 80.77%) and found a Hedges *g* effect of 1.79 (95%, PI [0.48 to 3.09]). Although, an inspection of the confidence intervals revealed that this did not differ significantly from studies that recorded treatment outcomes after six months (*g* = 1.66, 95% CI [1.15 to 2.17], *p* < 0.001, *I*^2^ = 85.30%), with a reported R^2^ value of 0.93%, indicating that treatment efficacy does not decrease after six months.

An additional subgroup analysis assessing psychedelic treatment concurrent with psychotherapy as a moderating factor was significant (*p* < 0.001, *I*^2^ = 80.40%), with a large effect size (*g* = 1.53, 95% PI [0.35 to 2.71]). Upon inspection of the confidence intervals, results revealed that psychedelic-assisted psychotherapy was not significantly different from studies employing psychedelic treatment alone (*g* = 1.98, 95% CI [1.53 to 2.43], *p* < 0.001, *I*^2^ = 82.95%), with an observed R^2^ of 11.43%, indicating that psychedelic treatment does not have to be combined with psychotherapy to be effective.

**Publication Bias.** Asymmetry due to publication bias for the combined studies was inspected through a funnel plot (see Figure A5). Visual inspection of the funnel plot depicts a moderately normal distribution of effect sizes. The Eggers regression test was non-significant (*p* = 0.128), and the Trim and Fill technique suggests that adding zero missing studies would correct asymmetry because of publication bias, suggesting minimal asymmetry and publication bias.

### 3.7. Risk of Bias Assessment

Out of 30 studies, 12 were RCTs, 16 were non-randomised or observational studies, and 2 were study protocols without outcome data. Most modern RCTs e.g., ref. [18,22] were rated with low risk in randomisation and missing data domains but had some concerns in deviations from intended interventions and outcome measurement. This is mainly due to lack of blinding and reliance on self-reported measures. Also, no RCTs were rated high risk overall. Many earlier or uncontrolled studies (e.g., early LSD trials and retrospective ibogaine surveys) were rated with serious risk, particularly due to confounding, small sample sizes, self-selection, and lack of blinding. Two protocol papers [42,76] were not rated. Overall, the evidence base combines several low-risk contemporary trials with multiple high-risk historical studies. This indicates that, while the pooled direction of effect favours psychedelic-assisted therapy, interpretation should be cautious given methodological heterogeneity and potential bias in a subset of studies.

## 4. Discussion

As a result of the war on drugs, research studies for psychedelic substances were banned in the early 1970s; therefore, the availability of current research is limited [77]. With the re-sparked interest around hallucinogens, the objective of this paper was to assess the efficacy of various psychedelic drugs as a treatment modality for SUD. The outcome yielded from the current findings suggests that psychedelics may assist in alleviating and abstaining from SUD. Secondary findings suggest that psychedelic therapy may be consistent over time and does not necessarily need to occur with psychotherapy to be an effective form of treatment to reduce SUD. This is the first meta-analysis investigating the difference and combined effectiveness between psilocybin, LSD, ketamine, and ibogaine for treating SUD. Thirty studies met the inclusion criteria, but six of these studies were excluded because the results could not be calculated into a standardised effect size [78,79,80] or the study assessed the effects of multiple psychedelics grouped together opposed to individual psychedelic effects [35]. One study in 2022 [39] reported promising effects of psilocybin in reducing opioid misuse but was excluded due to the results being reported as an odds ratio. Despite contacting the authors for further information, no response was provided. Most of the included studies had been published within the last decade, highlighting the revived interest surrounding psychedelic treatment.

The recent literature has similarly investigated the effects of psychedelics to treat SUD and SM [27,41,44]. Sharma et al. [44] employed a mixed-methods systematic review, which analysed qualitative and quantitative data. Zafar et al. [27] investigated the historical and futuristic perspective of psychedelic treatment for SM and Wong et al. [41] examined serotonergic psychedelics for SM.

A reoccurring pattern observed across the three studies was that individuals reported their psychedelic experience as significantly meaningful. Many individuals stated that these substances enabled them to gain insight into the root cause of their addictive behaviours. The introspection that followed psychedelic treatment often helped individuals understand the underlying and unresolved factors that initially contributed to the development of their SM.

Collectively, results indicated some therapeutic benefits of psychedelic treatment for SUD and SM but suggest that the evidence is insufficient and, therefore, unable to completely support the effectiveness of psychedelic treatment for SM. Although the researchers highlight the lack of evidence supporting psychedelic treatment, they demonstrate the need for future research.

As we explain below in detail, our results showed that psychedelics are an effective treatment for SM, while being efficacious over time. Importantly, compared to other psychedelics, ibogaine showed the strongest effect. A systematic review [81] expanded on the mechanisms behind psychedelic treatment for SM, suggesting that neuroplasticity could play a beneficial role. The authors pose that psychedelics may allow room for “therapeutic learning” through neuroplasticity, which can last far beyond a single dose. Neuroplasticity is the ability for the brain to re-wire itself, with psychedelic administration facilitating the process by altering brain network connectivity, emotional processing, reward and stress processing, social connectedness, and subjective experiences. When unmanaged, these factors may increase SM but, due to the impact psychedelic treatment may have on the brain’s neuroplasticity, could help further explain how it might be useful in reducing SM.

### 4.1. Summary of Results

The key findings from the 30 studies were analysed using a random-effects model, measuring pre-treatment to post-treatment efficacy for psilocybin, ketamine, and ibogaine. Due to the lack of information regarding participants’ baseline SUD, the efficacy of LSD could only be compared against each study’s control group. Consequently, the LSD studies could not be analysed against psilocybin, ketamine, and ibogaine. Therefore, the research question and aims were only partially fulfilled. Although LSD could not be compared to its counterparts, outcomes from the individual analysis depicted results favouring LSD as a treatment for SUD.

Analyses were conducted for psilocybin, ketamine, and ibogaine individually, with a subsequent analysis where the three psychedelics were combined. The outcomes exhibited statistically significant large effect sizes, indicating that psilocybin, ketamine, and ibogaine are successful forms of SUD management from baseline to post-treatment outcome.

When comparing the three individual analyses of psychedelic treatment groups, ibogaine produced the most significant effect, followed by ketamine and psilocybin. Upon observation of the combined psychedelic analyses, results revealed that a study assessing ibogaine for SM had the most significant effect size [70]. This signifies that ibogaine possesses the most significant influence on SUD reductions from pre-treatment to post-treatment outcomes.

Although ibogaine is classified as a hallucinogen, it differs from alternative psychedelic compounds that act as 5-HT2A agonists, like LSD and psilocybin. It has been suggested that the reported effects of ibogaine regarding opioid withdrawal do not involve much effect on 5-HT receptor [15,82]. Instead, ibogaine’s action may be complex, affecting multiple neurotransmitters collectively. Research suggests that it influences the NMDA, kappa-opioid, sigma, and nicotine receptors, with mechanisms interacting with the kappa-opioid receptors contributing to its psychoactive effects [17,83]. Ibogaine may work by reversing the effects of opiates on gene expression, and, as a result, neuroreceptors return to pre-addiction conditions. Despite the difference in psychedelic mechanisms, they have been thought to facilitate psychotherapeutic treatment [9,84].

Additionally, to expand on the collective analysis of ibogaine, psilocybin, and ketamine, two separate subgroup analyses were performed. The first subgroup analysis assessed time as a moderating factor, with post-treatment outcome recorded at less than six months compared to over six months. When observing the confidence intervals, a non-significant difference in short- and long-term efficacy for psychedelic treatment was revealed. Thus, treatment efficacy is consistent over time and may produce long-term benefits. Although LSD was not included in the combined psychedelic analysis, the current findings compared to the previous literature indicate that psilocybin, ketamine, and ibogaine may sustain long-term efficacy compared to LSD [26].

Moreover, Bogenschutz and Johnson, 2016 [12] suggested that further research should investigate the relationship between psychedelic efficacy and psychotherapy. Therefore, the second subgroup analysis assessed the relationship between psychedelic treatment and psychotherapy as a moderating factor. Results revealed a non-significant difference between psychedelic treatment concurrent with psychotherapy and psychedelic treatment alone, suggesting that psychedelic treatment does not need to concur with psychotherapy to be successful, therefore rejecting the prediction that psychedelic-assisted psychotherapy will have more significant benefits than isolated psychedelic treatment. These findings may contribute to future research on psychedelic treatment; psychotherapy can be expensive and emotionally demanding, which can deter individuals from participating [42,85]. Psychedelic treatment may appear more economically practical, less intimidating, and equally effective. Thus, individuals who sustain maladaptive SM behaviours may be more inclined to seek treatment.

In addition, the subgroup analyses on the combined psychedelic group provided valuable insights into the moderating effects of time and psychotherapy, but these findings must be interpreted with caution. The high heterogeneity observed in the individual ibogaine and ketamine groups suggests a high degree of variability that is not fully accounted for. Although the combined group’s larger sample size allowed us to perform these analyses, the results should be considered exploratory rather than conclusive. The limited number of primary studies in this field, particularly those with consistent methodologies, means that the precision and generalisability of our subgroup findings remain limited. Further detail regarding each psychedelic treatment is discussed below in the previous literature.

### 4.2. Psilocybin

A randomised clinical trial by Bogenschutz et al. [18] presented the highest effect size of the three included studies, suggesting that psilocybin is a successful treatment for alcoholism for up to 36 weeks. The current findings, simultaneous with the previous literature, corroborates the safety and efficacy of psilocybin [20,21]. Similar findings have been observed in tobacco addiction research [86], revealing that 12 out of 15 participants had ceased smoking at the 6-month follow-up subsequent to psilocybin treatment. Additionally, psilocybin has shown promising benefits in treating mental health conditions, including depression and anxiety [22,77,87]. Depression and anxiety commonly coincide with SM [20]; therefore, psilocybin treatment may assist in reducing SM while simultaneously improving individuals’ overall mental well-being.

Recent research studies further support these findings and expand the scope of psilocybin’s therapeutic potential for alcohol use disorder (AUD) and related conditions. A recent phase 2 randomised clinical trial found that psilocybin-assisted therapy was effective for relapse prevention in patients with AUD [22]. This study highlights the long-term benefits of psilocybin for sustaining recovery. Additionally, a pilot study using fMRI investigated how psilocybin affects neural reactivity in patients with AUD [21]. The study found that psilocybin-induced changes in brain activity in response to alcohol and emotional cues could be a key mechanism for its therapeutic effects. Ongoing research is also exploring psilocybin’s potential in other areas, with a recent protocol for a double-blind, randomised, placebo-controlled trial examining its efficacy for severe alcohol use disorder [76]. This study aims to evaluate the safety, feasibility, and clinical efficacy of psilocybin-assisted therapy during inpatient rehabilitation. Another study protocol has also been published for an open-label trial to examine the safety and efficacy of psilocybin-assisted therapy for veterans with PTSD, a condition that frequently co-occurs with substance misuse [42].

Research on psilocybin for SM is still limited; the final inclusion of studies was restricted due to the exclusion of case studies and studies containing qualitative data. Despite the group’s large effect size, psilocybin could not be extensively analysed and compared to ibogaine, ketamine, and LSD because only three studies were included. Nonetheless, the benefits of psilocybin treatment show hope for future SM treatment. To date, 33 international clinical trials for psilocybin have been completed, and 89 have been approved for treating addiction, mental health disorders, and other disorders [45], with clinical trials gaining approval in Australia in 2021.

### 4.3. Ketamine

Studies using ketamine as an SM treatment had the widest variety of misused substances, with efficacy observed for up to 24 months [36,66]. The most significant effect was observed in a study trialling ketamine-assisted behavioural treatment for cannabis use disorder [59]. Nonetheless, the current research suggests ketamine treatment may successfully reduce alcoholism, cocaine misuse, cannabis misuse, and heroin misuse. Further research is also trying to explore novel applications and mechanisms of ketamine for substance use disorders. One study repurposed ketamine to treat cocaine use disorder by integrating artificial-intelligence-based prediction, expert evaluation, and clinical corroboration [36]. This research identified ketamine as a promising candidate for treating this condition and provided insights into its potential mechanisms of action.

Ketamine is legalised and widely used in Australia for medical purposes. Doctors and veterinarians widely use ketamine as a pain killer and anaesthetic [17,88]. Additionally, ketamine is a successful treatment for mental health disorders, for example, treatment-resistant depression. These findings suggest that ketamine is advantageous over other psychedelics due to its frequent use by medical practitioners [89].

### 4.4. Ibogaine

Ibogaine maintains the most apparent effect size of all psychedelic treatments. A study assessing treatment outcomes for opioid dependence at the 12-month follow-up had the most significant effect [70], suggesting that ibogaine is effective over time. Most of the studies regarding ibogaine were employed to treat opioid misuse, with ibogaine exhibiting efficacy in reducing opioid addiction, craving, and withdrawals [15,43]. Preclinical trials in animal models of addiction have corroborated the effectiveness of ibogaine. Ibogaine has attenuated opioid withdrawals and self-administration of morphine, amphetamine, methamphetamine, nicotine, and ethanol in rodents and primates [17,90,91]. Therefore, future directions could apply ibogaine treatment for individuals who misuse previously stated substances.

Ibogaine is currently classed as a controlled substance in Australia, meaning it cannot be administered without a licence granted by the Therapeutic Goods Administration [92]. Nonetheless, this still poses an advantage for future treatment in Australia compared to the United States, where ibogaine is classed as a Schedule I substance. Therefore, all treatment exploration needs to be seriously addressed, as per a study from 2018 [90].

### 4.5. LSD

Findings from the group analysis favour LSD as an SM treatment. Bowen et al., 1970 [28] endeavoured to assess LSD as a variable in the treatment of alcoholism, which presented the most significant effect of all studies. Thus, LSD may be most appropriate for treating alcohol misuse.

Conversely, most LSD research ended in 1970, not long after Ludwig et al., 1969 [30] and Hollister, 1969 [29] conducted thorough evaluations of LSD treatment for SM and concluded no advantage over oppositional treatments. Moreover, Abuzzzbah and Anderson (1971) [93] indicated inconclusive results about the efficacy of LSD for SM. Despite past research suggesting the inconclusive efficacy of LSD, conflicting findings suggest otherwise. Newer research has corroborated the efficacy of LSD for heroin use disorder, anxiety, depression, psychosomatic illnesses, and anxiety due to life-threatening diseases for up to 12 months, proposing that individuals may benefit from LSD treatment [94]. Nonetheless, LSD is classified as a schedule nine substance in Australia; therefore, possessing and selling this drug is illegal. However, LSD is not illegal for medical research.

### 4.6. Limitations

By synthesising the results of numerous studies, a more reliable representation of the findings can be produced, which cannot be as easily achieved by evaluating individual primary studies [95]. Individual studies comprised small sample sizes, but a larger sample size is produced by assessing studies collectively and, therefore, statistical power is increased. Additionally, using a repeated-measures design allows researchers to determine how effective treatment is from baseline to post-treatment outcome.

In contrast to measuring the difference between post-treatment outcomes to the control opposition, a precise representation of drug efficacy may not necessarily be depicted. For example, a treatment and control outcome comparison does not inform the reader how much improvement was made from baseline to post-treatment, only the discrepancy between the treatment and control. However, the current repeated-measures meta-analysis violated the assumption that data for all individuals was accessible, potentially introducing bias [96]. A future recommendation involves exploring the current research aim using an independent-measure meta-analysis.

An advantage of using a repeated-measures methodology is that it also controls for placebo effects. Placebo effects in clinical control trials can be fundamentally flawed, as they can potentially introduce the invalid determination of causal inferences [97]. However, in the LSD treatment studies, an accurate determination of baseline to post-treatment change could not be established and placebo effects could not be controlled. The lack of sufficient data presented in the LSD studies could be attributable to the experiments conducted over 20 years ago, inferring those results may be outdated. Therefore, future research could expand on LSD clinical trials for SM by comparing post-treatment efficacy to participant baseline SM.

Nevertheless, the current research has some limitations. Firstly, the research question and article could not be fulfilled entirely. This is due to the LSD studies using a different methodological analysis, which compared the post-treatment outcome to the studies’ control comparator. Therefore, LSD could not be appropriately compared to alternative psychedelics as a treatment for SM. The final studies included were smaller than expected, especially for psilocybin. Consequently, the psilocybin subgroup was disadvantaged, as it could not be extensively analysed against ketamine, ibogaine, and LSD, which all included more than five studies. Future research could employ investigation and synthesisation of the case studies and qualitative papers trialling psilocybin as an SM treatment. Secondly, the assessment of methodological quality of the included studies was not extensively reported, therefore introducing the risk of bias, thus threatening the current review’s internal validity [98].

This meta-analysis faced a methodological limitation related to the comparative designs of the included studies. The LSD literature, comprised of older studies, primarily used a treatment vs. control design. In contrast, many of the more recent psilocybin, ketamine, and ibogaine studies were based on a pre-treatment to post-treatment design. While we addressed this by performing separate analyses, this distinction complicates direct cross-intervention comparisons. A significant methodological limitation was the considerable variability observed within the individual ibogaine and ketamine treatment groups. The combined effect sizes for psilocybin, ketamine, and ibogaine, while significant, are based on a different comparative framework than the LSD studies. Therefore, these results should not be interpreted as a direct comparison of efficacy between the older and newer psychedelic compounds but, rather, as distinct findings that reflect the evolving methodologies within the field of psychedelic research. The limited number of primary studies in this emerging field, particularly those with consistent methodologies, suggests that the generalisability of these findings may be limited. The subgroup analyses on time and psychotherapy should therefore be viewed as exploratory rather than definitive, highlighting the need for more primary research to confirm these effects.

Additional analyses due to heterogenous results were not investigated for individual psychedelic interventions, only for the combined psychedelic treatment interventions. Further analyses, namely, subgroup analyses, help differentiate qualitative and quantitative interactions and aim to identify consistency or significant differences among treatment effects [99]. However, sub-analyses could not be conducted due to the limited sample size for individual psychedelic groups [75]. Therefore, the influence of other variables on the results could not be explained. Although this may present a limitation, conflicting literature suggests that this should not be of concern, as subgroup analyses can be misleading [100].

Psychedelic substances also possess negative stigmatisation, although this has not always been true. In the 1950s and 1960s, the government supported and funded psychedelic research. However, psychedelic use became intertwined with “hippy culture”. Recreational use, concurrent with the war on drugs, may have contributed to the root of psychedelic stigmatisation [101]. Even though psychedelic research is underway in Australia, there remains speculation about therapeutic journeys still being conducted in unregulated conditions by therapists with minimal to no qualifications [101]. The motivation to do so could be influenced by a quest to obtain therapeutic potential restricted and denied by current legislation [102,103,104]. Psychedelic treatment conducted in unregulated conditions, with no certainty of purified product or patient screening, can be hazardous. Although psychedelics have displayed remarkable outcomes, it cannot be assumed that all hallucinogens will affect people similarly, especially in uncontrolled conditions. Thus, another reason for future research is to investigate the safe and controlled administration of psychedelic treatment.

## 5. Conclusions

The main objective of this meta-analysis was to investigate if psychedelic treatment could successively alleviate and reduce SUD. Outcomes of the current paper suggested that psychedelic treatment for SUD possesses many benefits, particularly given the limited efficacy of existing treatments [12,104]. Each psychedelic analysed in the current article demonstrates adequacy as an SUD treatment. Additional findings reflect psychedelic efficacy for up to two years, suggesting that psychedelic treatment may not always require concurrent psychotherapy to be successful. However, due to the heterogeneity and differences in study methodologies, these findings should be interpreted with caution and require further investigation. Psilocybin, LSD, ketamine, and ibogaine demonstrated efficacy as a treatment, with ibogaine attaining the most prominent effect. The current findings suggest that future research should consider a further investigation into longitudinal studies examining psychedelic treatment for SUD. It seems unethical to withhold the exploration of significant new findings that can lead to valuable treatment outcomes for those who need it the most.

## 6. Implication

This systematic review and meta-analysis offers a consolidated look at four different psychedelic agents as potential treatments for substance use disorder (SUD). While many past studies have focused on one psychedelic at a time, our work brings them together, which helps us see the bigger picture. This unique perspective provides some important insights for both future clinical practice and research.

Our findings can help clinicians and researchers think about which psychedelics might be most promising. For instance, ibogaine showed the strongest effect in our analysis, but we know it also carries serious cardiac risks. This highlights the need to weigh both a drug’s potential benefits and its safety profile very carefully. This study suggests that the therapeutic efficacy of psychedelic interventions is not solely attributable to the drug’s acute effects but, in addition, is likely mediated by the concurrent role of psychotherapeutic support, highlighting the need for future research to elucidate this synergistic relationship.

For the research community, our work suggests some clear paths forward. By comparing these four agents, we can better prioritise which ones deserve more funding and larger clinical trials. We need more rigorous studies on drugs like LSD, which have a long history but lack new data. We also need future research to build on our findings about the role of psychotherapy and the longevity of treatment effects. This can help us design better, more effective treatment programmes for those who need them most.

## Figures and Tables

**Figure 1 healthcare-13-02668-f001:**
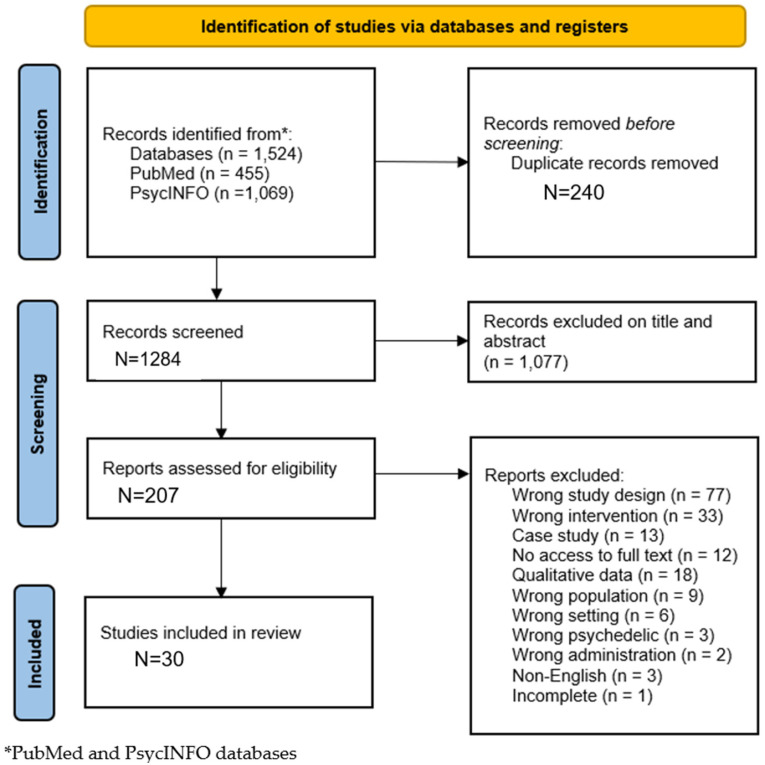
PRISMA version 2020 flow diagram [47] for a meta-analysis investigating the efficacy of various psychedelic drugs for the treatment of substance misuse.

**Figure 2 healthcare-13-02668-f002:**
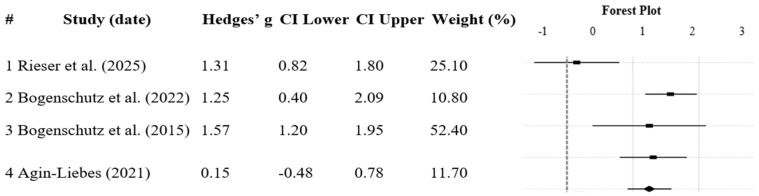
Forest plot depicting psilocybin efficacy for substance misuse [18,22,24,58]. Note: Black dots with lines spanning left to right represent individual studies’ effect size and confidence intervals; the bottom line represents the overall effect size and confidence interval. The # shows serial number.

**Figure 3 healthcare-13-02668-f003:**
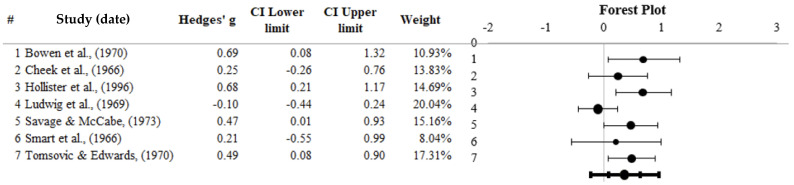
Forest plot representing LSD and control for substance misuse treatment [10,28,29,32,33,34,74]. Note: Black dots spanning to the right are effect sizes in favour of the LSD condition; the bold line represents the overall effect size and confidence interval. The # shows serial number.

**Figure 4 healthcare-13-02668-f004:**
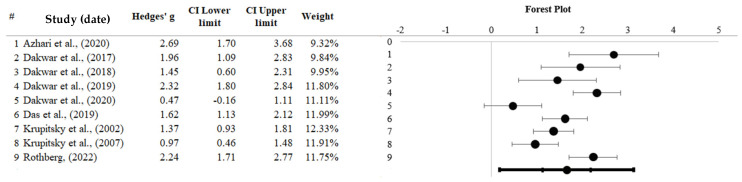
Forest plot representing ketamine efficacy for substance misuse [38,59,60,61,62,63,65,66,67]. Note: Black dots represent individual study effect sizes, the left and right lines represent the confidence intervals, and the bottom line represents the overall effect size and confidence interval. # shows to a serial number.

**Figure 5 healthcare-13-02668-f005:**
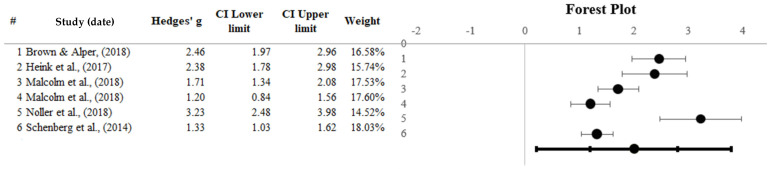
Forest plot representing ibogaine effect sizes [43,64,68,69,70]. Note: Black dots represent individual study effect sizes, the left and right lines represent the confidence intervals, and the bottom line represents the overall effect size and confidence interval. # shows to serial number.

**Figure 6 healthcare-13-02668-f006:**
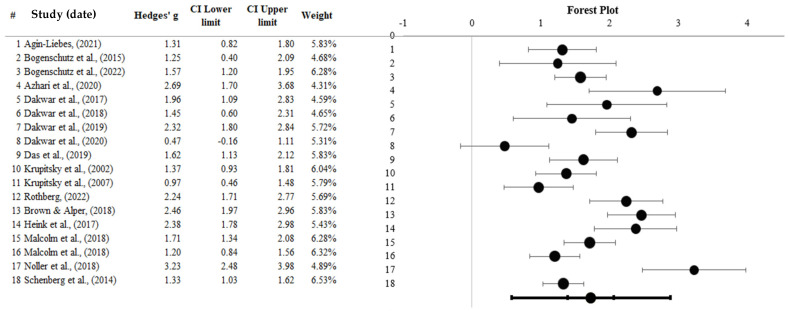
Forest plot representing psilocybin, ketamine, and ibogaine studies [18,24,38,43,58,59,60,61,62,63,64,65,66,67,68,69,70]. Note: Black dots represent individual study effect sizes, the left and right lines represent the confidence intervals, and the bottom line represents the combined groups’ overall effect size and confidence interval. # shows to serial number.

## Data Availability

There is no data associated with this study, as it is a meta-analysis.

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
