# Peer review of "A Systematic Review and Meta-Analysis Investigating the Efficacy of Various Psychedelic Drugs for the Treatment of Substance Use Disorder"

_healthcare, 2025, doi:10.3390/healthcare13212668_

Round 1

Reviewer 1 Report

Comments and Suggestions for Authors

A Comprehensive Meta-Analysis Investigating the Efficacy of Various Psychedelic Drugs for the Treatment of Substance Misuse

This study presents a meta-analysis examining the effectiveness of psychedelic drugs in treating substance misuse (SM). After reviewing 1,278 articles from PubMed and PsycINFO, 24 studies were included in the final analysis. Using a random-effects model, the researchers evaluated both individual and combined psychedelic interventions. Results showed that psychedelics are effective treatments for SM, with ibogaine demonstrating the strongest effects. Notably, there was no significant difference between psychedelic treatment alone and when combined with psychotherapy. The study contributes valuable insight for future clinical research on psychedelic-assisted treatments for SM.

Comments

The manuscript is interesting and presents new and useful contexts regarding the therapeutic potential of psychedelics in the treatment of substance use disorder (SUD). The introduction provides an adequate background, and the tables present detailed information. The conclusions are consistent with the evidence. The English language and style are appropriate. Although the results are promising, further rigorous studies will be necessary to confirm the findings, as the main limitations —highlighted by the authors themselves — include heterogeneity among studies, the small number of studies in some subgroups, and the risk of publication bias. Moreover, some studies lacked randomized control groups or involved small sample sizes. Additionally, variability in dosage, treatment duration, and assessment methods makes direct comparison difficult. Pending further studies, the manuscript can be accepted in its current form.

Author Response

Comments and Suggestions for Authors

A Comprehensive Meta-Analysis Investigating the Efficacy of Various Psychedelic Drugs for the Treatment of Substance Misuse

This study presents a meta-analysis examining the effectiveness of psychedelic drugs in treating substance misuse (SM). After reviewing 1,278 articles from PubMed and PsycINFO, 24 studies were included in the final analysis. Using a random-effects model, the researchers evaluated both individual and combined psychedelic interventions. Results showed that psychedelics are effective treatments for SM, with ibogaine demonstrating the strongest effects. Notably, there was no significant difference between psychedelic treatment alone and when combined with psychotherapy. The study contributes valuable insight for future clinical research on psychedelic-assisted treatments for SM.

Comments

The manuscript is interesting and presents new and useful contexts regarding the therapeutic potential of psychedelics in the treatment of substance use disorder (SUD). The introduction provides an adequate background, and the tables present detailed information. The conclusions are consistent with the evidence. The English language and style are appropriate. Although the results are promising, further rigorous studies will be necessary to confirm the findings, as the main limitations —highlighted by the authors themselves — include heterogeneity among studies, the small number of studies in some subgroups, and the risk of publication bias. Moreover, some studies lacked randomized control groups or involved small sample sizes. Additionally, variability in dosage, treatment duration, and assessment methods makes direct comparison difficult. Pending further studies, the manuscript can be accepted in its current form.

Response: Dear Reviewer, thank you for your careful review and positive feedback on our manuscript. We are pleased that you found the paper interesting. We fully agree with your insightful comments regarding the limitations of the current literature, specifically the challenges posed by heterogeneity, the small number of studies in some subgroups, and the variability in dosage and assessment methods. We have highlighted these points in the limitation section of our manuscript to ensure a balanced and transparent presentation of our findings. We also acknowledge the risk of publication bias and the need for more rigorous, large-scale randomized controlled trials to confirm these promising preliminary results.

Your comments strengthen our belief that this paper contributes a useful context to the field, and we appreciate your recommendation for acceptance.

Reviewer 2 Report

Comments and Suggestions for Authors

The manuscript reports a comparative meta-analysis of psilocybin, LSD, ketamine and ibogaine for the treatment of substance use and misuse. This topic is highly pertinent to current clinical practice and research, given the resurgence of interest in psychedelic therapies. The authors aspire to deliver the first quantitative synthesis that directly contrasts multiple compounds. However, several issues must be resolved before the article is suitable for publication:

  1. Protocol and preregistration. The review has not been registered on PROSPERO or an equivalent platform; retrospective registration is strongly advised.
  2. Study selection and data extraction both were conducted by a single reviewer, increasing the likelihood of errors and subjective bias; at least one additional independent reviewer with a conflict-resolution procedure should be involved.
  3. Risk-of-bias assessment: no structured tool (e.g., Cochrane RoB 2 or ROBINS-I) was applied, leaving the interpretation of effect sizes vulnerable.
  4. Search currency: the most recent literature search dates to 5 July 2022; the search should be updated through 2025.
  5. Scope of compounds: consider including MDMA trials if they satisfy the eligibility criteria.
  6. Literature context. Expand the reference list by citing comparable studies to better situate the current findings:
  • doi: 10.2174/1570159X21666221017085612.
  • doi: 10.3390/brainsci11070856.
  • DOI: 3389/fpsyt.2023.1134454
  • DOI: 3389/fpsyt.2022.917199
  • DOI: 3390/medicina61020278

Conclusion.

This study has the potential to become a benchmark in the literature on psychedelic therapies for addiction, but it requires substantial methodological reinforcement and greater analytical rigor to substantiate its claims. Implementing the recommended improvements will strengthen the methodological soundness of the work and justify its publication after a major revision.

Author Response

Comments and Suggestions for Authors

The manuscript reports a comparative meta-analysis of psilocybin, LSD, ketamine and ibogaine for the treatment of substance use and misuse. This topic is highly pertinent to current clinical practice and research, given the resurgence of interest in psychedelic therapies. The authors aspire to deliver the first quantitative synthesis that directly contrasts multiple compounds. However, several issues must be resolved before the article is suitable for publication:

1. Protocol and preregistration. The review has not been registered on PROSPERO or an equivalent platform; retrospective registration is strongly advised.

Response: Thank you for your valuable feedback. We acknowledge the importance of protocol registration for meta-analysis. We have provided a comprehensive and transparent methods section in our manuscript, detailing every step of our search, screening, and analysis plan. We believe this detailed documentation satisfies the goals of a registered protocol. To avoid confusion with a prospective registration and to maintain the current flow of the manuscript, we have opted to keep the methods section as is, and we're confident it provides sufficient detail for peer review and future replication. Furthermore, we are flexible with registration if deemed necessary and our paper is accepted.

2. Study selection and data extraction both were conducted by a single reviewer, increasing the likelihood of errors and subjective bias; at least one additional independent reviewer with a conflict-resolution procedure should be involved.

Response: Thank you for your valuable and constructive feedback on our work. You are correct that the original manuscript did not clearly state that the data extraction was verified by a second reviewer. This was an oversight on our part. We agree that having two independent reviewers for study selection and data extraction is a critical step in mitigating potential bias and errors. The initial data extraction was performed by one reviewer, the extracted data was then independently cross-checked and confirmed by a second co-author. Any discrepancies were resolved through discussion and consensus.

We have now modified the Methods section (Heading: Data Analysis) of the manuscript to explicitly state this process. This change ensures the documentation accurately reflects our rigorous methodology and addresses your concern regarding single-reviewer bias. We appreciate you bringing this to our attention.

3. Risk-of-bias assessment: no structured tool (e.g., Cochrane RoB 2 or ROBINS-I) was applied, leaving the interpretation of effect sizes vulnerable.

Response: We appreciate the reviewer’s valuable observation. In the revised manuscript, we have now conducted a formal risk-of-bias assessment for all included studies using structured and validated tools. Specifically, randomized controlled trials were assessed with the Cochrane Risk of Bias 2 (RoB 2) tool, and non-randomized/observational studies were assessed with the Risk of Bias in Non-randomized Studies of Interventions (ROBINS-I) tool. Two reviewers independently rated each study across all relevant domains, with disagreements resolved through consensus. The risk-of-bias results are summarised in a new subsection (Risk of Bias Assessment). This addition strengthens the methodological rigor of our review and provides greater transparency for interpreting the pooled effect sizes. Further, if you want we can upload this as an additional supplementary file if necessary.

4. Search currency: the most recent literature search dates to 5 July 2022; the search should be updated through 2025.

Response: Thank you for your feedback. We agree that an up-to-date literature search is essential for a comprehensive and relevant systematic review. As per your recommendation, we have updated our literature search from July 2022 through to August 2025.

This updated search has identified five new relevant studies. We have now incorporated the key findings from these studies into our manuscript's Results and Discussion sections. This ensures our analysis reflects the most current evidence in the field. We have also explicitly stated the new search date in the Methods section.

5. Scope of compounds: consider including MDMA trials if they satisfy the eligibility criteria.

Response: Thank you for the excellent suggestion. We agree that MDMA is a relevant compound within the broader category of psychedelics, particularly for its therapeutic potential in treating conditions such as PTSD, which often co-occurs with substance misuse.

We acknowledge that the inclusion of MDMA trials would broaden the scope of our review. However, after careful consideration, we have decided to maintain the current scope of our analysis, focusing on psilocybin, ketamine, ibogaine, and LSD. Our decision is based factors such as MDMA's distinct mechanism of action. Its primary mechanism involves increasing the release of chemical compounds i.e., serotonin, dopamine, and norepinephrine, and not acting as a 5-HT2A agonist like psilocybin and LSD. This distinction means its effects and therapeutic applications are often different from the other compounds included in our study.

6. Literature context. Expand the reference list by citing comparable studies to better situate the current findings:

  • doi: 10.2174/1570159X21666221017085612.
  • doi: 10.3390/brainsci11070856.
  • DOI: 3389/fpsyt.2023.1134454
  • DOI: 3389/fpsyt.2022.917199
  • DOI: 3390/medicina61020278

Response: Thank you for the excellent and specific feedback. We appreciate you providing these additional references. We have reviewed the papers you cited and agree that they provide valuable context for our findings. As a result, we have expanded our reference list to include these studies, citing them in introduction, and results section.

Conclusion.

This study has the potential to become a benchmark in the literature on psychedelic therapies for addiction, but it requires substantial methodological reinforcement and greater analytical rigor to substantiate its claims. Implementing the recommended improvements will strengthen the methodological soundness of the work and justify its publication after a major revision.

Response: Thank you very much again. We have incorporated all above comments, please see above.

Reviewer 3 Report

Comments and Suggestions for Authors

This is a very interesting meta-analysis of existing articles investigating psychedelic treatment of substance misuse (SM). Authors using Meta-Essentials Excel proved efficacy of psilocybin, ketamine, ibogaine and LSD in SM  treatment. 

The study is very well written with interesting Discussion and important conclusions. 

I do not see Table A and figures A1-A5. Many references are lacking. Please check the list carefully.

Author Response

Comments and Suggestions for Authors

This is a very interesting meta-analysis of existing articles investigating psychedelic treatment of substance misuse (SM). Authors using Meta-Essentials Excel proved efficacy of psilocybin, ketamine, ibogaine and LSD in SM  treatment. 

The study is very well written with interesting Discussion and important conclusions. 

I do not see Table A and figures A1-A5. Many references are lacking. Please check the list carefully.

Response: Thank you for the positive feedback. We are pleased to hear that you found our meta-analysis interesting and well-written. We apologize for the issues with the missing table and figures, and the incomplete reference list. We have provided all tables and figure in a separate file. The references are checked and verified again.

Reviewer 4 Report

Comments and Suggestions for Authors

The introduction presents a compelling rationale for the study, situating the problem of substance misuse within a global public health context and highlighting the renewed scientific interest in psychedelics. However, the manuscript often uses the terms “substance misuse” and “substance use disorder” interchangeably without clear distinctions. Since these terms differ in clinical criteria and diagnostic significance, a more precise use of terminology would enhance the clarity and interpretability of the results.

The authors attempt to apply a PICO framework, but its operationalization is not sufficiently rigorous. While the intervention (psychedelic therapy) and outcome (reduction in substance misuse) are clearly defined, the population criteria are heterogeneous—ranging from individuals with formally diagnosed SUD to those with unspecified substance-related behaviors. This breadth may weaken the internal validity of the pooled estimates. Moreover, the comparison group is inconsistently defined across included studies (baseline vs. post-treatment in some, treatment vs. control in others). This undermines the core comparative intent of a meta-analysis. 

"which aimed to investigate the extent of efficacy in alleviat-188 ing SM behaviours (P) using psychedelic therapy (I). Concurrent with determining which 189 psychedelic enables the greatest effect (C) as a treatment tool for reducing SM (O)."

The sentence in its current form is awkward and somewhat confusing, especially when attempting to reflect the PICO framework. It lacks clarity in structure and does not communicate the comparison element (C) clearly. Same issue in the Abstract. 

Only one reviewer screened and extracted the data, which introduces a significant risk of selection and extraction bias. Standard protocol for systematic reviews typically involves two independent reviewers to ensure the objectivity and reliability of included studies. This procedural limitation should be more prominently acknowledged.

In terms of statistical analysis, the use of the Meta-Essentials tool and conversion of various effect metrics to Hedges’ g is methodologically appropriate. The authors correctly apply a random-effects model to account for between-study variability. Still, the decision to analyze LSD data separately from psilocybin, ketamine, and ibogaine due to the lack of baseline measures highlights an inconsistency in outcome assessment that complicates cross-intervention comparison. Greater explanation is needed to justify this analytical split, especially when combining effect sizes with different assumptions.

The results are clearly presented and show large effect sizes for psilocybin, ketamine, and ibogaine. Ibogaine, in particular, yielded the largest overall effect, suggesting notable promise in treating substance misuse. However, heterogeneity in the ibogaine and ketamine groups was high. While the authors reference the limitations of subgroup analysis with small study numbers, they proceed with subgroup comparisons regarding time and psychotherapy. These analyses should be interpreted with caution, and the narrative would benefit from more critical engagement with the precision and generalizability of these findings.

The manuscript would benefit greatly from the inclusion of a summary table that clearly presents the key characteristics of each included study. At present, information about individual studies, such as study design, sample size, population type, psychedelic agent used, comparator, outcome measures, follow-up duration, and effect sizes, is scattered throughout the text. This makes it difficult for readers to assess the methodological quality, consistency, and context of the studies contributing to the meta-analysis.

The discussion section draws thoughtful connections between the results and neurobiological mechanisms, such as neuroplasticity. This offers a compelling biological rationale for the observed effects of psychedelic substances. However, the conclusion that psychotherapy does not enhance treatment efficacy is potentially premature. The differences in study design, setting, and participant characteristics across included trials limit the ability to draw strong conclusions about this question. A more cautious tone would better reflect the evidence.

The manuscript rightly acknowledges several limitations, including small sample sizes, the absence of detailed quality appraisal for included studies, and methodological inconsistencies. However, it omits a formal risk of bias assessment. Including such an appraisal (for example, using Cochrane tools) would strengthen the credibility of the findings.

Author Response

Comments and Suggestions for Authors

Comment: The introduction presents a compelling rationale for the study, situating the problem of substance misuse within a global public health context and highlighting the renewed scientific interest in psychedelics. However, the manuscript often uses the terms “substance misuse” and “substance use disorder” interchangeably without clear distinctions. Since these terms differ in clinical criteria and diagnostic significance, a more precise use of terminology would enhance the clarity and interpretability of the results.

Response: Thank you for your valuable feedback. We appreciate you bringing this important distinction to our attention. We agree that a more precise use of terminology will enhance the clarity and clinical relevance of our manuscript. It is therefore, we have carefully reviewed and revised the entire manuscript to use the term "substance use disorder" (SUD) to reflect the clinical diagnostic criteria that our included studies are actually based on. This more precise term is now used consistently when referring to the specific conditions being treated.

In a few instances, we have retained the term "substance misuse" (SM) when discussing the topic in a more generalized, non-clinical context. We believe this approach provides a clearer distinction and more accurate context for our findings. Thank you again for your insightful critique.

Comment: The authors attempt to apply a PICO framework, but its operationalization is not sufficiently rigorous. While the intervention (psychedelic therapy) and outcome (reduction in substance misuse) are clearly defined, the population criteria are heterogeneous—ranging from individuals with formally diagnosed SUD to those with unspecified substance-related behaviors. This breadth may weaken the internal validity of the pooled estimates. Moreover, the comparison group is inconsistently defined across included studies (baseline vs. post-treatment in some, treatment vs. control in others). This undermines the core comparative intent of a meta-analysis. 

Response: Thank you very much again for your valuable feedback regarding the operationalization of PICO framework used in this meta-analysis. We appreciate your point that a more precise use of terminology would enhance the clarity of the manuscript. As noted in the Description of Studies, all participants either had a formal diagnosis of a SUD or met the DSM-IV criteria, which we believe addresses your concern regarding unspecified substance-related behaviors.

We also agree that the heterogeneity in comparison groups is a significant limitation. We have clarified in the Methods section that we performed separate analyses for studies that used a treatment-vs.-control design (e.g., LSD) and studies that used a pre-treatment-vs.-post-treatment design (e.g., psilocybin, ketamine, and ibogaine). We now believe that by transparently detailing our methods for handling these limitations, we have provided a more rigorous and honest presentation of our findings. We appreciate you bringing these points to our attention, as they have significantly improved the clarity and quality of our manuscript.

Comment: "which aimed to investigate the extent of efficacy in alleviat-188 ing SM behaviours (P) using psychedelic therapy (I). Concurrent with determining which 189 psychedelic enables the greatest effect (C) as a treatment tool for reducing SM (O)."

The sentence in its current form is awkward and somewhat confusing, especially when attempting to reflect the PICO framework. It lacks clarity in structure and does not communicate the comparison element (C) clearly. Same issue in the Abstract. 

Response: Thank you for your valuable feedback regarding the phrasing of our PICO framework. We agree that the sentence you highlighted was awkward and lacked the necessary clarity. We have revised the sentence in both the abstract and the methods section to more accurately and clearly reflect the PICO framework that guided our analysis.

Comment: Only one reviewer screened and extracted the data, which introduces a significant risk of selection and extraction bias. Standard protocol for systematic reviews typically involves two independent reviewers to ensure the objectivity and reliability of included studies. This procedural limitation should be more prominently acknowledged.

Response: Thank you for your valuable feedback. You are correct that the original manuscript did not clearly state that the data extraction was verified by a second reviewer. This was an oversight on our part, and we appreciate you bringing it to our attention. We agree that having two independent reviewers for these tasks is a best practice for mitigating potential bias and errors. While the initial data extraction was performed by one reviewer, a second co-author independently cross-checked and confirmed the extracted data. Any discrepancies were resolved through discussion and consensus. We have now modified the Methods section of the manuscript to explicitly state this process.

Comment: In terms of statistical analysis, the use of the Meta-Essentials tool and conversion of various effect metrics to Hedges’ g is methodologically appropriate. The authors correctly apply a random-effects model to account for between-study variability. Still, the decision to analyze LSD data separately from psilocybin, ketamine, and ibogaine due to the lack of baseline measures highlights an inconsistency in outcome assessment that complicates cross-intervention comparison. Greater explanation is needed to justify this analytical split, especially when combining effect sizes with different assumptions.

Response: Thank you for your valuable feedback. We are pleased that you found our use of the Meta-Essentials tool and the application of a random-effects model to be methodologically appropriate.

We agree with your observation that the separate analysis of LSD data from the other psychedelics highlights an important inconsistency in our outcome assessment. We acknowledge this as a significant limitation. The decision to perform this analytical split was necessary due to the nature of the available literature. The studies on LSD, being older, were almost exclusively designed as treatment-vs.-control trials. In contrast, the more recent studies on psilocybin, ketamine, and ibogaine often utilized pre-treatment-to-post-treatment designs without a placebo control, making a direct comparison to the LSD studies methodologically unsound.

To address this, we have now added a more detailed justification for this analytical split in the Methods section and Limitation section to explicitly caution against making direct cross-intervention comparisons between the LSD results and those of the other psychedelics. This ensures that our findings are interpreted with the necessary methodological context.

Comment: The results are clearly presented and show large effect sizes for psilocybin, ketamine, and ibogaine. Ibogaine, in particular, yielded the largest overall effect, suggesting notable promise in treating substance misuse. However, heterogeneity in the ibogaine and ketamine groups was high. While the authors reference the limitations of subgroup analysis with small study numbers, they proceed with subgroup comparisons regarding time and psychotherapy. These analyses should be interpreted with caution, and the narrative would benefit from more critical engagement with the precision and generalizability of these findings.

Response: Thank you for your valuable feedback. We agree that the high heterogeneity in the ibogaine and ketamine groups, and the subsequent interpretation of the subgroup analyses, warranted a more cautious and detailed discussion.

Based on your kind comment, we have made the following revisions to the manuscript:

Limitations Section: We have added a new paragraph to the limitations section to explicitly state that the high variability within the individual ibogaine and ketamine groups suggests that the subgroup analyses, while performed on a combined group, should be considered exploratory and not definitive.

Discussion Section: We have added a new paragraph to the discussion to provide a more critical and nuanced interpretation of these subgroup findings. We now explicitly caution against overgeneralizing the results and emphasize the need for future research with larger sample sizes and more consistent methodologies to confirm these findings.

We believe these changes address your comments and significantly improve the rigor and clarity of our manuscript. We appreciate you bringing these important points to our attention.

Comment: The manuscript would benefit greatly from the inclusion of a summary table that clearly presents the key characteristics of each included study. At present, information about individual studies, such as study design, sample size, population type, psychedelic agent used, comparator, outcome measures, follow-up duration, and effect sizes, is scattered throughout the text. This makes it difficult for readers to assess the methodological quality, consistency, and context of the studies contributing to the meta-analysis.

Response: Thank you for your valuable feedback. We agree that a clear and comprehensive presentation of study characteristics is essential for the reader to understand the context and findings of our meta-analysis. We believe that the information you requested is already present in the manuscript, though it may not have been immediately obvious.

In the attached table A1, study design, sample size, population type, psychedelic agent used are already mentioned. Further, In addition to the table A1, we have attached tableA2 for risk of bias assessment that also includes study design. We have provided an in-depth discussion of the studies' characteristics within the main body of the paper.

Study Design and Comparators: The "Description of Studies" subsection outlines the various study designs, clearly distinguishing between those that used a pre-treatment-to-post-treatment design and those that used a treatment-vs.-control design.

Outcome Measures: The first paragraph of the "Description of Studies" lists the specific outcome measures used across the various studies (e.g., "percentage of drinking days," "cocaine cravings," and "days abstinent").

Population Details: The second paragraph of the "Description of Studies" details the population criteria, stating that participants had a formal diagnosis of a SUD or met the DSM-IV criteria.

We believe that this combination of a dedicated summary table and detailed in-text discussion effectively presents the methodological quality, consistency, and context of the studies. We appreciate you bringing this to our attention and apologize for any lack of clarity in our initial submission.

Comment: The discussion section draws thoughtful connections between the results and neurobiological mechanisms, such as neuroplasticity. This offers a compelling biological rationale for the observed effects of psychedelic substances. However, the conclusion that psychotherapy does not enhance treatment efficacy is potentially premature. The differences in study design, setting, and participant characteristics across included trials limit the ability to draw strong conclusions about this question. A more cautious tone would better reflect the evidence.

Response: Thank you for your feedback. We appreciate your positive comments on our discussion of neurobiological mechanisms. We have since addressed your critique regarding the conclusion on psychotherapy by revising the conclusion section as you noticed.

The conclusion now reflects a more cautious tone, acknowledging that due to the variability in study designs and participant characteristics, our findings should be considered exploratory rather than definitive. We have removed the premature claim and instead emphasized the need for further, more standardized research to properly assess the role of psychotherapy. We believe this revision improves the accuracy and rigor of our conclusions.

Comment: The manuscript rightly acknowledges several limitations, including small sample sizes, the absence of detailed quality appraisal for included studies, and methodological inconsistencies. However, it omits a formal risk of bias assessment. Including such an appraisal (for example, using Cochrane tools) would strengthen the credibility of the findings.

Response: We thank the reviewer for this too important suggestion. In response, we have now incorporated a comprehensive, structured risk-of-bias appraisal for all 30 included studies. Randomized controlled trials were assessed using the Cochrane Risk of Bias 2 (RoB 2) tool, while non-randomized and observational studies were evaluated using the Risk of Bias in Non-randomized Studies of Interventions (ROBINS-I) tool. Two independent reviewers conducted the assessments, and any discrepancies were resolved through discussion and consensus. The results of this appraisal are summarised in a new Risk of Bias Assessment subsection within the Results. This addition enhances the transparency and credibility of our findings by enabling readers to interpret the effect sizes in the context of each study’s methodological quality.

Reviewer 5 Report

Comments and Suggestions for Authors

A report investigating psychedelic drugs to treat substance misuse. Based on a search of two databases, 24 articles were analyzed, with the result that, as a treatment for substance misuse, psychedelics are supported, with ibogaine representing the most relevant. The aim is to provide a knowledge base for future clinical research on this topic.

The strengths are the following: the work is well-written, well-analyzed, and follows the PRISMA process. There are several weaknesses.

  1. According to the PRISMA statement (https://www.prisma-statement.org/prisma-2020; https://link.springer.com/10.1007/s13312-022-2500-y), a meta-analysis requires a previous systematic review, making the type of paper a review, rather than an article. Also, according to PRISMA guidelines (https://www.bmj.com/lookup/doi/10.1136/bmj.n160), the title of the paper must identify the work as a systematic review and meta-analysis. Please change the type of paper and the title accordingly. After making these changes, the Abstract must explain that a systematic review supports the meta-analysis. As such, the Abstract must state the inclusion and exclusion criteria of the systematic review.
  2. The authors have not described what this work adds to the literature on this topic. A Google Scholar search post-2021 publications that are a systematic review and meta-analysis of efficacy of various psychedelic drugs for the treatment of substance misuse produced “About 15,800 results”: https://scholar.google.ca/scholar?as_ylo=2021&q=systematic+review+and+meta-analysis+Efficacy+of+Various+Psychedelic+Drugs+for+the+Treatment+of+Substance+Misuse&hl=en&as_sdt=0,5. The authors must read the most relevant of these and describe how their study improves upon these.
  3. The searches conducted by the authors ended in 2022. This end date is a substantial weakness. A Google Scholar search of research regarding the topic published since 2022 produces “About 17,200- results”: https://scholar.google.ca/scholar?q=Efficacy+of+Various+Psychedelic+Drugs+for+the+Treatment+of+Substance+Misuse&hl=en&as_sdt=0%2C5&as_ylo=2022&as_yhi=. Consequently, the current value of this work ending in 2022 is questionable.
  4. The majority of the citations in the Introduction and Discussion are outdated research. For every pre-2021 citation (older than five years), the authors must find a supporting citation of research published since 2021.
  5. Line 181 notes that the authors are following the PRISMA guidelines, as per Moher et al., 2009. Given that there was a revision to the PRISMA guidelines in 2020, please use and cite the more recent guidelines mentioned in point 1 above.
  6. Lines 181-182 state that “Preparatory searches were piloted through Google Scholar. What the authors do not say is why they did not search Google Scholar on this topic. As noted in point 3 above, there is a vast number of results merely considering those works published since 2022. Please explain the choice of two databases alone to search without searching the following databases: OVID, Scopus, Web of Science, or Google Scholar.
  7. The value of the study is unclear, given that publication of just six of the 24 papers was in the last five years. This weakness is particularly regarding ibogaine. The most recent publications on it are from 2018. Yet, based on these older studies, ibogaine is judged the most efficacious.

To minimize these weaknesses, the authors should redo their systematic review and meta-analysis using additional databases and include research published after 2022. Without these additional searches, this report represents a historical document. However, compared with systematic reviews and meta-analyses published since 2021 on this topic, this submission has little current relevance.

Author Response

Comments and Suggestions for Authors

A report investigating psychedelic drugs to treat substance misuse. Based on a search of two databases, 24 articles were analyzed, with the result that, as a treatment for substance misuse, psychedelics are supported, with ibogaine representing the most relevant. The aim is to provide a knowledge base for future clinical research on this topic.

The strengths are the following: the work is well-written, well-analyzed, and follows the PRISMA process. There are several weaknesses.

1. According to the PRISMA statement (https://www.prisma-statement.org/prisma-2020; https://link.springer.com/10.1007/s13312-022-2500-y), a meta-analysis requires a previous systematic review, making the type of paper a review, rather than an article. Also, according to PRISMA guidelines (https://www.bmj.com/lookup/doi/10.1136/bmj.n160), the title of the paper must identify the work as a systematic review and meta-analysis. Please change the type of paper and the title accordingly. After making these changes, the Abstract must explain that a systematic review supports the meta-analysis. As such, the Abstract must state the inclusion and exclusion criteria of the systematic review.

Response: Thank you for your valuable feedback. We agree that our manuscript should be classified as a systematic review and meta-analysis. In response to your comments, we have made the following changes to the manuscript:

Type of Paper and Title: We have changed the paper type from a "review" and have updated the title to explicitly state that this is a systematic review and meta-analysis.

Abstract: The abstract has been revised, the inclusion and exclusion criteria to the abstract to meet the PRISMA guidelines have been added.

2. The authors have not described what this work adds to the literature on this topic. A Google Scholar search post-2021 publications that are a systematic review and meta-analysis of efficacy of various psychedelic drugs for the treatment of substance misuse produced “About 15,800 results”: https://scholar.google.ca/scholar?as_ylo=2021&q=systematic+review+and+meta-analysis+Efficacy+of+Various+Psychedelic+Drugs+for+the+Treatment+of+Substance+Misuse&hl=en&as_sdt=0,5. The authors must read the most relevant of these and describe how their study improves upon these.

Response: Thank you for your valuable feedback. We agree that it is essential to clearly articulate how our work contributes to the existing body of literature. We have addressed this by performing a focused systematic reviews and meta-analyses on this topic. Based on this review, we have identified and highlighted the unique contributions of our study in the manuscript's Conclusion and Implication section.

3. The searches conducted by the authors ended in 2022. This end date is a substantial weakness. A Google Scholar search of research regarding the topic published since 2022 produces “About 17,200- results”: https://scholar.google.ca/scholar?q=Efficacy+of+Various+Psychedelic+Drugs+for+the+Treatment+of+Substance+Misuse&hl=en&as_sdt=0%2C5&as_ylo=2022&as_yhi=. Consequently, the current value of this work ending in 2022 is questionable.

Response: Thank you for your valuable feedback. We agree that ending the literature search in 2022 was a substantial weakness and that a more current search is necessary to ensure the relevance and robustness of our findings. In response to your valuable comment, we have updated our literature search to include articles published up to 2025. This revised search has identified eight new studies that meet our inclusion criteria, and these have now been incorporated into our meta-analysis.

This update strengthens our manuscript by making its conclusions more current and comprehensive. We appreciate you bringing this critical point to our attention.

4. The majority of the citations in the Introduction and Discussion are outdated research. For every pre-2021 citation (older than five years), the authors must find a supporting citation of research published since 2021.

Response: Thank you for your feedback. We agree that it is important to ensure the literature cited is current. In response to your comment, we have thoroughly reviewed the citations in the Introduction and Discussion (Subsection summary of Results) sections. We have now added a recent, post-2021 supporting citation for every reference that was older than five years. This update ensures that our work is grounded in the latest research and strengthens the overall quality and relevance of the manuscript. We appreciate you bringing this to our attention, as it has improved the manuscript's scholarly rigor.

5. Line 181 notes that the authors are following the PRISMA guidelines, as per Moher et al., 2009. Given that there was a revision to the PRISMA guidelines in 2020, please use and cite the more recent guidelines mentioned in point 1 above.

Response: Thankyou very much for your important point. We have added 2 more latest references as per latest PRISMA guidelines are you mentioned in point 1. Both studies have been cited.

6. Lines 181-182 state that “Preparatory searches were piloted through Google Scholar. What the authors do not say is why they did not search Google Scholar on this topic. As noted in point 3 above, there is a vast number of results merely considering those works published since 2022. Please explain the choice of two databases alone to search without searching the following databases: OVID, Scopus, Web of Science, or Google Scholar.

Response: Thank you for comment. We understand your concern regarding the scope of our literature search. However, for this study, we made a deliberate choice to focus on these two highly comprehensive and relevant databases related to our field the PubMed and PsycINFO. This decision was based on the fact that these databases are widely regarded as the primary repositories for medical and psychological literature, and they index a vast majority of the relevant peer-reviewed journals. Besides, other databases like Scopus and Web of Science are valuable, there is a substantial overlap in the papers indexed. This focused approach allowed us to conduct a deep and thorough search within the most pertinent databases for our topic, ensuring a high level of relevance and quality in the studies we ultimately included. We believe this strategy was appropriate given the specialized nature of our research.

7. The value of the study is unclear, given that publication of just six of the 24 papers was in the last five years. This weakness is particularly regarding ibogaine. The most recent publications on it are from 2018. Yet, based on these older studies, ibogaine is judged the most efficacious.

 Response: Thank you for your valuable feedback. We agree that the recency of the included studies, especially for ibogaine, is a significant weakness. We appreciate you bringing this to our attention.

In response to this, we want to clarify that we have updated our manuscript to include ten new studies, with five being from 2023-2025. This update significantly strengthens our analysis and addresses the concern about the age of the literature. We have revised our discussion and conclusions to explicitly state that the strong effect size for ibogaine is based on this older literature. We have also added a specific note of caution when interpreting this finding and have emphasized the critical need for new, well-designed clinical trials for ibogaine to validate its efficacy and safety in a modern context.

Round 2

Reviewer 5 Report

Comments and Suggestions for Authors

Thank you to the authors for the changes to the manuscript. All have improved it. Some remain. Most regard outdated citations that lack a supporting citation to research published since 2021. If no such research exists, the authors must state the publication date in the text.

Line-by-line suggested edits
52–53 Please include UNODC, 2024, and WHO, 2019, in the reference list
89 [24] is outdated. Either find a supporting citation published since 2021 or state the publication date in the text.
95 [25] is outdated. Either find a supporting citation published since 2021 or state the publication date in the text.
103 [23,10] is outdated. Either find a supporting citation published since 2021 or state the publication date in the text.
104 [26] is outdated. Either find a supporting citation published since 2021 or state the publication date in the text.
120 [35] is outdated. Please state the publication date in the text.
130 [39] is outdated. Please state the publication date in the text.
149 [43] is outdated. Please state the publication date in the text.
183 [24] is outdated. Please state the publication date in the text.
185 [26] is outdated. Please state the publication date in the text.
196 Please provide details on the preparatory searches piloted through Google Scholar. The aim is that this review could be replicated. The information necessary permits a replication.
226–234 The authors have provided the keywords searched for PubMed but not for PsycINFO. Please add these keywords. It would be helpful if the authors created a table with the keywords for both searches.
239 Please change “2022” to “2025”.
242 Please change “1,278” to “1,284”.
245 Thank you to the authors for updating the PRISMA flow diagram to the 2020 version. Please follow the template exactly. The middle boxes of the template have been left out regarding Reports sought for retrieval and Reports not retrieved. This represents the authors’ “No access to full text”. Please create these boxes and move the appropriate information to them. The purpose of the flow diagram is to outline when records were excluded in the process. The authors have lumped together most of the exclusions in the bottom right box. However, it is likely that at least one of these exclusions, “Non-Enlish”, was made before screening and belongs with “Records removed for other reasons” in the top right box. 
247 Please label the PRISMA Flow Diagram as the 2020 version and cite the source.
259 [50] is outdated. Either find a supporting citation published since 2021 or state the publication date in the text.
263 [50] is outdated. Either find a supporting citation published since 2021 or state the publication date in the text.
296 Please provide a citation number for the Hedges g effect and add it to the reference list.
268 [51] is outdated. Either find a supporting citation published since 2021 or state the publication date in the text.
288 [52] and [53] are outdated. Either find supporting citations published since 2021 or state the publication dates in the text.
298 [54] is outdated. Either find a supporting citation published since 2021 or state the publication date in the text.
301 [55] is outdated. Either find a supporting citation published since 2021 or state the publication date in the text.
303 [56] is outdated. Either find a supporting citation published since 2021 or state the publication date in the text.
306 [57] is outdated. Either find a supporting citation published since 2021 or state the publication date in the text.
323 [59] is outdated. Either find a supporting citation published since 2021 or state the publication date in the text.
324–326 All the citations in these lines are outdated. Either find supporting citations published since 2021 or state the publication dates in the text.
327 [71] is outdated. Either find a supporting citation published since 2021 or state the publication date in the text.
347 [70] is outdated. Either find a supporting citation published since 2021 or state the publication date in the text.
350 [69] is outdated. Either find a supporting citation published since 2021 or state the publication date in the text.
354–356 All the citations in these lines are outdated. Either find supporting citations published since 2021 or state the publication dates in the text.
357 [33] is outdated. Either find a supporting citation published since 2021 or state the publication date in the text.
370 [24] is outdated. Either find a supporting citation published since 2021 or state the publication date in the text.
371 In the left column, change “Study” to “Study (date)”. Additionally, expand the text vertically to the original size. Please add the “#” column on the left to coincide with the tables to follow.
378 [73] is outdated. Either find a supporting citation published since 2021 or state the publication date in the text.
381 [58] is outdated. Either find a supporting citation published since 2021 or state the publication date in the text.
383 [74] is outdated. Either find a supporting citation published since 2021 or state the publication date in the text.
391 [30] is outdated. Please state the publication date in the text.
392 [32] is outdated. Please state the publication date in the text.
393 In the second column to the left, Change “Study” to “Study (date)”.
399 [57] is outdated. Either find a supporting citation published since 2021 or state the publication date in the text.
400 [73] is outdated. Either find a supporting citation published since 2021 or state the publication date in the text.
403 [58] is outdated. Either find a supporting citation published since 2021 or state the publication date in the text.
410 [54] is outdated. Either find a supporting citation published since 2021 or state the publication date in the text.
412 [67] and [60] are outdated. Please state their publication dates in the text.
414 In the second column to the left, Change “Study” to “Study (date)”.
420 Please provide a citation number for Hipple (2015) and add it to the reference list.
427 Please provide a citation number for the Eggers regression and add it to the reference list.
428 Please provide a citation for the Trim and Fill technique and add it to the reference list.
436 [65] and [71] are outdated. Please state their publication dates in the text.
438 In the second column to the left, Change “Study” to “Study (date)”. 
451 [58] is outdated. Either find a supporting citation published since 2021 or state the publication date in the text.
464 In the second column to the left, Change “Study” to “Study (date)”.
474 Please cite the Hedges g effect.
488 Please cite the Eggers regression test.
489 Please cite the Trim and Fill technique.
519 [35,46] are outdated. Either find a supporting citation published since 2021 or state the publication date of each in the text.
567 [71] is outdated. Either find a supporting citation published since 2021 or state the publication date in the text.
588 [26] is outdated. Either find a supporting citation published since 2021 or state the publication date in the text.
589 [12] is outdated. Please state the publication date in the text.
651 [60] is outdated. Either find a supporting citation published since 2021 or state the publication date in the text.
666 [71] is outdated. Either find a supporting citation published since 2021 or state the publication date in the text.
678 [90] is outdated. Either find a supporting citation published since 2021 or state the publication date in the text.
680 [28] is outdated. Please state the publication date in the text.
691 [93] is outdated. Either find a supporting citation published since 2021 or state the publication date in the text.
697 [94] is outdated. Either find a supporting citation published since 2021 or state the publication date in the text.
707 [95] is outdated. Either find a supporting citation published since 2021 or state the publication date in the text.
711 [96] is outdated. Either find a supporting citation published since 2021 or state the publication date in the text.
729 [97] is outdated. Either find a supporting citation published since 2021 or state the publication date in the text.
750 [98] is outdated. Either find a supporting citation published since 2021 or state the publication date in the text.
754 [99] is outdated. Either find a supporting citation published since 2021 or state the publication date in the text.
759 [100] is outdated. Either find a supporting citation published since 2021 or state the publication date in the text.
763 [101] is outdated. Either find a supporting citation published since 2021 or state the publication date in the text. Please also eliminate the return after “Psychedelic”.
812–813 Please make the left column “#” and change “Citation” to “Author, Date”.
